# Comparison of Submillimeter Spot Ablation of Copper and Nickel by Multipulse Picosecond and Femtosecond Laser

**Mingyu Li [1], Jifei Ye [1,\*], Lan Li [1], Bangdeng Du [2], Ying Wang [1], Heyan Gao [1] and Chenghao Yu [1]**

1   State Key Laboratory of Laser Propulsion & Application, Space Engineering University, Beijing 101416, China
2   Technical Institute of Physics and Chemistry, Chinese Academy of Sciences, Beijing 100190, China
\*   Correspondence: yjf1981@163.com

**Abstract:** The current transmission and reflection laser ablation micropropulsion modes have the problem of a complex working medium supply system in engineering. Therefore, we propose large-spot laser ablation with a one-dimensional supply mode. In order to verify this ablation mode, a multipulse ablation experiment of submillimeter-scale light spots was carried out on the surface of pretreated copper and nickel under the atmosphere using an ultrafast laser with a pulse width of 290 fs and 10 ps. The results show that femtosecond laser multipulse ablation (FLMA) leads to the grain refinement of copper, the crater quality of the two metals under FLMA is better, and picosecond laser multipulse ablation (PLMA) causes the crater of nickel to form a dense remelting bulge that affects laser absorption; both metals have obvious heat-affected zones after FLMA and PLMA, the heat-affected zones of nickel are 5–10% larger than those of copper, and the ablation depth of copper is deeper. Under the same conditions, the ablation mass of copper is smaller than that of nickel, and the specific impulse performance of laser ablation micropropulsion is better.

**Keywords:** laser ablation micropropulsion; grain refinement; ablation morphology; ablation depth; ablation mass; specific impulse





## 1. Introduction

Ultrafast laser has the characteristics of ultrafast time domain and ultra-high peak power, which is one of the important tools in the frontier of basic scientific research, and the interaction between ultrafast laser and matter is one of the directions [1,2]. The most typical application field of ultrafast laser interaction with matter is ultrafast laser processing [3]. The advantages of noncontact and cold processing make it one of the most advanced manufacturing technologies [4]. Ultrafast laser acts on the material surface after focusing, and intense energy injection in an ultrashort time will produce high-temperature and high-pressure vapor, which can quickly achieve material removal, and accompanied by impulse generation, it can be used in precision laser processing, laser ablation propulsion, laser removal of space debris, and other fields [5–8].

Surface ablation morphology, ablation depth, and ablation mass are important parameters of laser ablation metal, and they are important indexes to evaluate the quality of laser processing of metal materials and the propulsion performance of laser ablation metal [9–11]. The factors that determine the above indexes are ablation materials, ablation environment, laser characteristics, and so on [10,12,13]. For practical laser ablation microthrusters, the thrust is usually adjusted by adjusting the laser characteristics [14]. At present, the main pulse widths for precision laser ablation metal processing are ps and fs [3,15]. Under the same conditions, the surface quality of FLMA is smoother, and the ablation depth of different materials irradiated by the same laser is also different [15–17]. The removal efficiency of ultrashort laser at 1064 nm wavelength is higher than that at short wavelength [18]. Repetition rate also has an effect on ablation efficiency [19]. At a fixed repetition rate, the thermal effect of PLMA is stronger at low flux; when the flux is high, the thermal effects of

both are obvious [17]. Single-pulse fs laser irradiates metal with deeper ablation depth than ps laser [20]. The plasma induced by ultrafast laser ablation is affected by the number of laser pulses [21,22]. The study of multipulse ablation shows that the ablation mass plays a key role in evaluating the processing efficiency of laser processing and the specific impulse of laser ablation propulsion [23], but the ablation mass of ultrafast laser is small, and it is difficult to measure it accurately. The volume method is a more accurate method [11].

At present, research on ultrafast laser ablation metal propulsion mainly focuses on a single pulse and a small number of pulses, and there are few studies on multipulse (>100) laser ablation propulsion. In the actual application of laser ablation propulsion, ablation with multiple pulse numbers is necessary, but the laser light source is generally fixed, which leads to the problem of a continuous supply of propulsion working medium. In the existing several types of laser ablation propulsion prototypes, the mode of small spot-focused ablation (transmission mode and reflection mode) is adopted [23,24], and a complex working medium supply system is equipped for continuous thrust output. In this paper, we propose the idea of laser ablation advancement with a one-dimensional working medium supply. In the one-dimensional supply mode, a large spot is used for ablation, which can facilitate the preparation of the grain working medium, and can also obtain a larger mass propellant at a certain length.

In the experiment, we use ultrafast lasers with two pulse widths of 290 fs and 10 ps to study the comparison of ablation of copper and nickel with different energies and laser pulse numbers at a fixed repetition rate, and measured and analyzed the surface ablation morphology, depth, and quality of the material, to verify the feasibility of the one-dimensional supply mode of large-spot laser ablation.

## 2. Experimental Equipment and Materials

In this study, the PHAROSPH2-10W (PH2) laser system produced by LIGHTCONVER-SION Company (located in Vilnius Lithuania) was used as the ultrafast laser pulse source. The adjustable parameters of PH2 included pulse duration (290 fs–20 ps), repetition rate (up to 1 MHz), pulse energy (up to 400 μJ), and average power (up to 10W). Other parameters of the laser included central wavelength (1030±10 nm) and beam quality (TEM$_{00}$, M$^2$ < 1.2). The pulsed laser energy in the experiment was measured by the Newport thermoelectric sensor. The experimental materials were copper and nickel round-metal sheets with a diameter of 10 mm and thickness of 1 mm, and their purity was 99.9%. Before the experiment, the surface of the sample was polished with a grinding and polishing machine. The surface of the sample was successively polished with 3000 mesh, 1500 mesh, and 800 mesh sandpaper for 3–5 min, and then the sample was polished with a flannel coated with a polishing paste. Finally, the roughness measurement instrument was used to control the range of Ra of the sample surface roughness to be 0.2–0.4 μm. The schematic diagram of the experimental setting is shown in Figure 1. In the atmospheric environment, PH2 was controlled by special software to generate a fixed frequency of a 1 KHz multipulse laser beam. The parameters of the laser beam are shown in Table 1.

**Table 1.** Laser parameters in the experiment.

| Laser Pulse Width | Single Pulse Laser Energy/μJ | Number of Laser Pulses | | | | |
|---|---|---|---|---|---|---|
| | 100 | 200 | 400 | 600 | 800 | 1000 |
| 290 fs10 ps | 150 | 200 | 400 | 600 | 800 | 1000 |
| | 200 | 200 | 400 | 600 | 800 | 1000 |

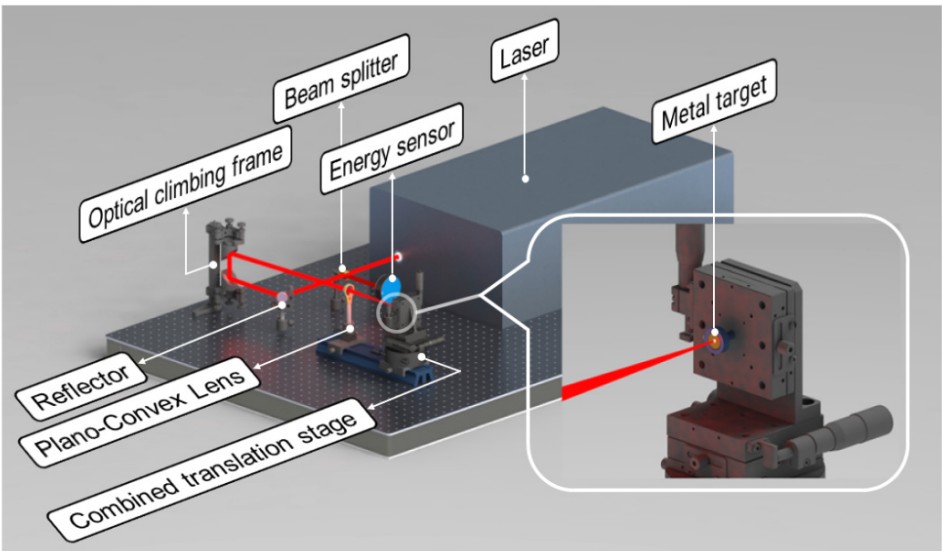

**Figure 1.** Schematic diagram of the laser ablation experimental setup.

Through three steering lenses with reflectivity greater than 99.6%, including the optical climbing frame, the laser beam deflected the optical path by 90 degrees and lifted the 200 mm. The laser beam finally passed through a flat-convex lens with 660–1380 nm antireflection coating and the focal length f = 100 mm and irradiated the surface of the metal target. In the experiment, the distance between the target and the lens was kept unchanged. In order to achieve submillimeter light spots, the target was located at 1 mm in front of the lens focus, and the combined translation stage was used to control the distance between the ablation points to be 2 mm. The ZEISS Axiolab5 microscope (Carl Zeiss AG, Oberkochen, Germany), the FEI QUANTA FEG250 field emission scanning electron microscope, and Olympus OLS5100 laser confocal microscopy (Olympus Corporation, Tokyo, Japan) were used to observe and measure the diameter, surface ablation morphology, and ablation volume of the crater.

## 3. Results and Discussion

### 3.1. Laser Confocal Microscopy Observation Results

3.1.1. Surface Ablation Morphology Observation Results

The ablation morphology and ablation depth of copper and nickel surfaces were observed by laser confocal microscopy. The photo of the surface ablation morphology of the targets is shown in Figure 2. It can be seen from the group pictures that the surface morphology of the targets after multipulse ablation of copper and nickel by ultra-short-pulse laser can be divided into two regions, namely, the crater region in the middle and the ablation product region in the outer layer. It can be seen from the crater regions that both copper and nickel have incomplete ablation or adhesion of ablation products, but their manifestations are different. The color of the crater surface of copper is darker, and there is uniform ablation product adhesion, while the surface of the craters of nickel has remelting bulge substances. Then, observe the ablation product regions of the targets under different working conditions. It can be seen from the lateral comparison of the group pictures that with the increase in the incident laser energy, the radius of the annular ablation product regions outside the copper crater gradually increases. When the incident laser energy is small, the adhesion of the ablation products is thin, which cannot completely cover some machining ravines on the copper surface. When the incident laser energy is 200 µJ, there is no phenomenon that the metal surface is not covered in the ablation product region. However, the ablation product regions of nickel do not show obvious regularity, which is related to the remelted bulge substances formed in its crater.

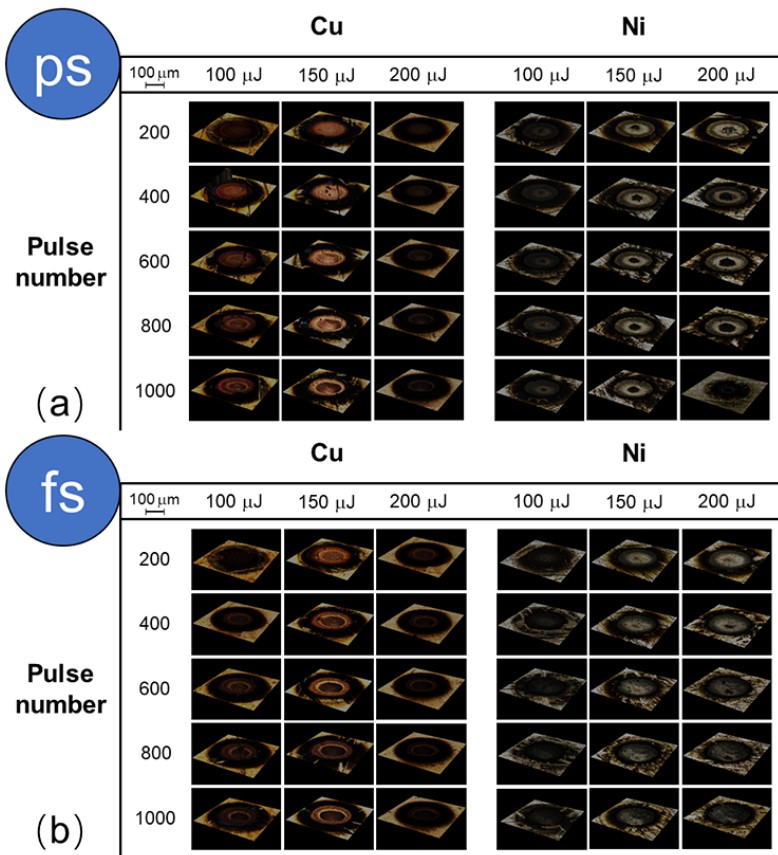

**Figure 2.** Surface ablation morphologies obtained by Laser confocal microscopy for Ni and Cu specimens under different pulse energies and number of pulses for: (**a**) PLMA; (**b**) FLMA.

### 3.1.2. Three-Dimensional Ablation Depth Measurement Results

As can be seen from the ablation depth group pictures in Figure 3 for the targets, the red line is the measured path through the deepest point of the craters automatically scanned by the software. The positions of the deepest points of the craters for the ablation of copper by the two pulse-width lasers have good consistency, and most of them are distributed in the center of the craters. The deepest points of nickel craters are unevenly distributed due to the presence of remelted bulge substances. In contrast, the quality of nickel craters ablated by fs laser will be smoother. In addition, we can also see the height of the accumulation of ablation products around the craters of copper and nickel. The height of the accumulation products around the copper craters is more uniform, while the ablation products around the nickel craters are scattered, which is consistent with what is observed in the surface ablation morphology pictures.

### 3.2. Scanning Electron Microscope Observation Results

The crater morphology of two metals ablated by ultrafast laser with 1000 pulses was characterized by field emission environmental scanning electron microscopy, as shown in Figure 4. It can be seen from the group pictures that under the same laser pulse width, the diameter of the crater of laser ablation of copper is smaller than that of nickel, but the crater is flat. Remelted bulge substances in the craters of ps laser-ablated nickel show a dense pore-like structure under electron microscopic observation.

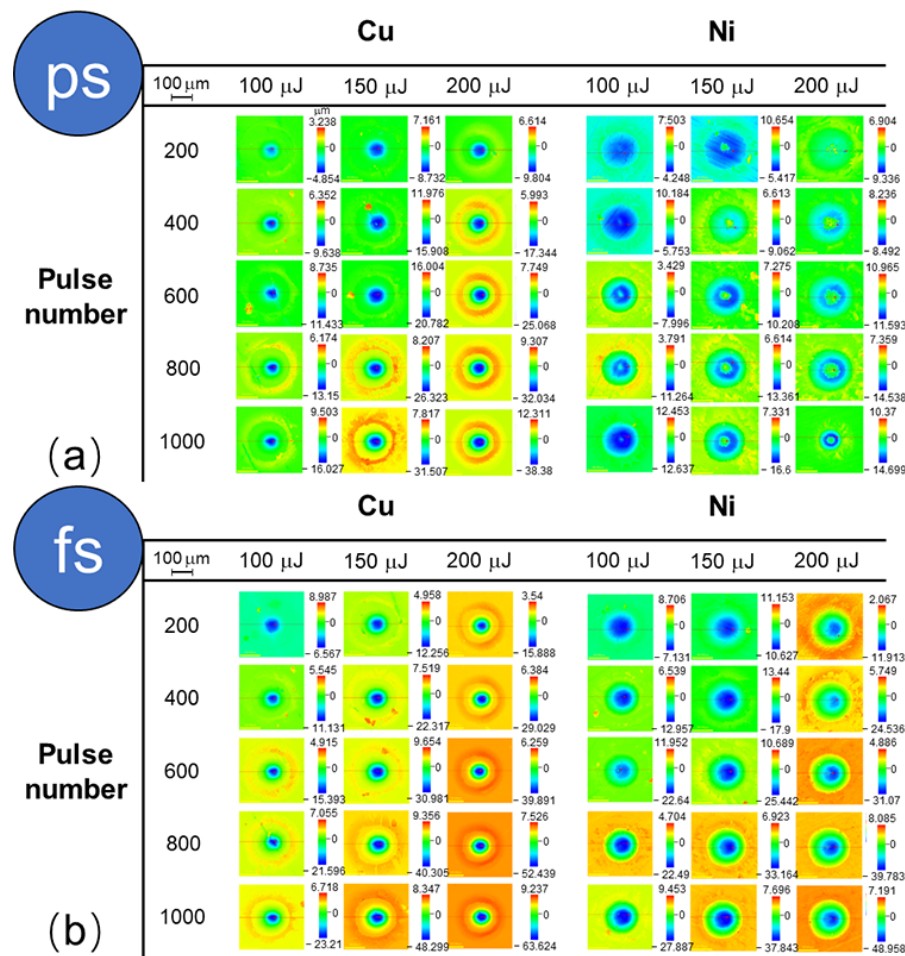

**Figure 3.** 3D profiles obtained by laser confocal microscopy for Ni and Cu specimens under different pulse energies and number of pulses for: (**a**) PLMA; (**b**) FLMA.

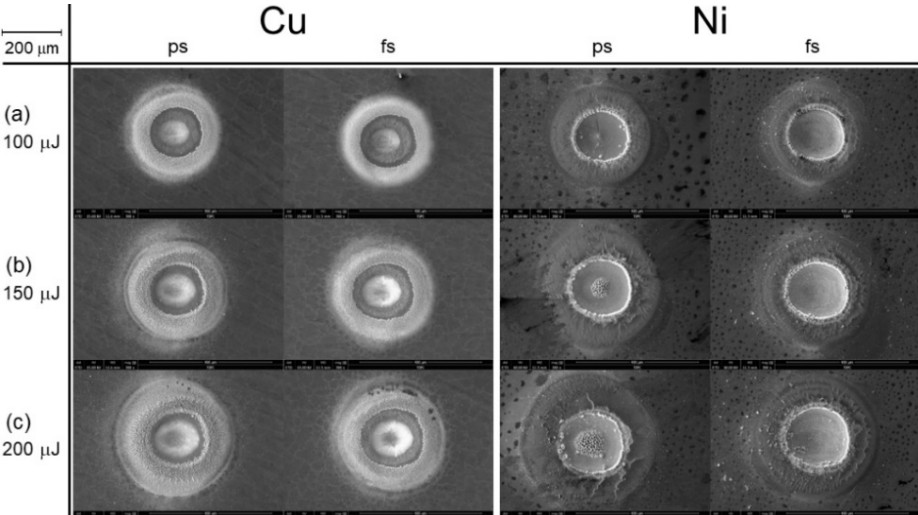

**Figure 4.** Surface ablation morphologies obtained by SEM for Ni and Cu specimens under different pulse widths and same number of pulses (1000) for: (**a**) 100 µJ; (**b**) 150 µJ; (**c**) 200 µJ.

### 3.3. Surface Ablation Morphology Analysis

Figure 5 shows the surface ablation morphology of copper observed with different instruments under experimental conditions of 290 fs, 200 μJ, and fixed 1000 pulses, which can be generally divided into three regions.

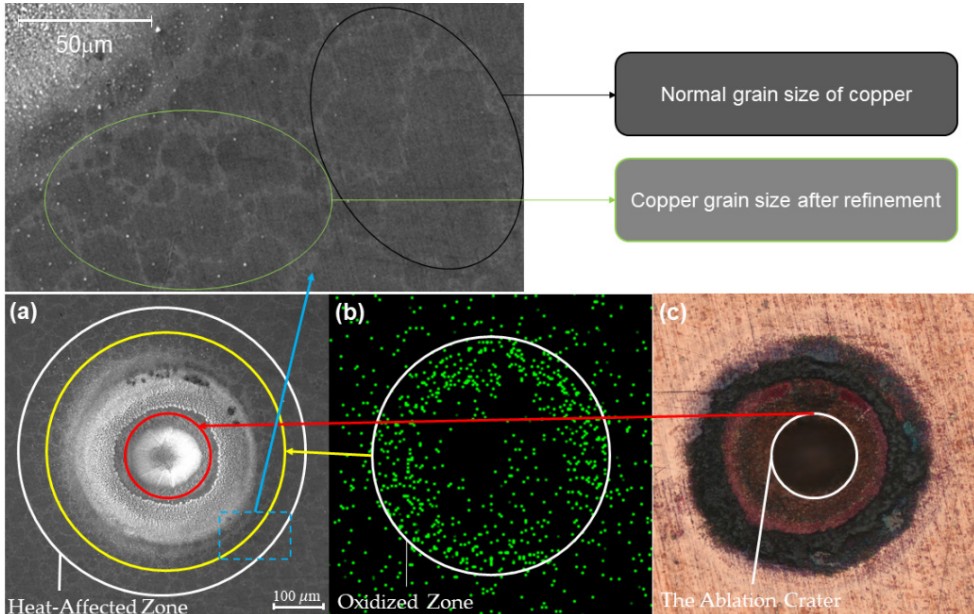

**Figure 5.** Surface ablation morphology of copper after laser irradiation with energy of 200 μJ, pulse width of 290 fs and pulse number of 1000 ((**a**) is SEM image, (**b**) is EDS analysis of oxygen element, (**c**) is optical microscopy photo).

As can be seen from the SEM picture (Figure 5a), there are metal recrystallized grains on the periphery of the crater, which are smaller and denser than the normal metal grains outside the region. This is due to the cumulative diffusion of the thermal effect of high repetition frequency multipulse laser ablation on the metal grains, which leads to the grain refinement of copper, which can effectively prevent dislocations and intergranular slippage and increase the pull-up intensity. Improve mechanical properties such as toughness. This part can be seen as the maximum range of thermal damage area caused by laser irradiation of metal, which is the first kind of surface topography heat-affected zone (HAZ). Through the EDS oxygen element analysis (Figure 5b), it can be observed that there is an obvious distribution of oxygen atoms inside the HAZ, which is shown as a black annular area in Figure 5c, which is caused by the high-temperature gradient of ultrashort pulse laser ablation. The closer to the ablation crater, the higher the temperature, and the metal molecules are active and prone to oxidation. This is the second kind of region-oxidation zone. In the center of the whole ablation area is the location of the ablation crater, that is, the third kind of area, the ablation crater area, with a submillimeter diameter.

Figure 6 shows the variation of three kinds of surface ablation topography of copper and nickel under the condition of 1000 pulses but with different laser energy and pulse width. By comparison, it can be seen that under the same experimental conditions, the HAZ of the target after the action of ps laser is about 5% to 10% higher than that of fs laser.

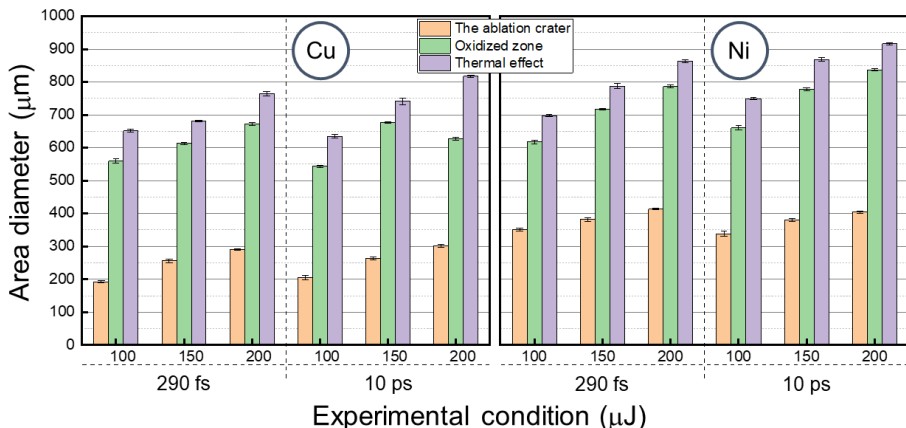

**Figure 6.** Regional variations in surface morphology of laser ablation of copper and nickel with different energies and pulse widths for 1000 pulse numbers.

According to the two-temperature equation of the interaction between fs laser and metal proposed by Anisimov [25], fs laser ablation of metal is a kind of nonequilibrium ablation, which is divided into two different processes: photon and electron, and electron and lattice. The former is usually in the range of fs, while the latter is usually in the order of ps, so when the metal is ablated by ps laser, the laser energy is transmitted to the lattice when the metal is irradiated by laser pulse, so the thermal effect will be stronger.

$$
\begin{aligned}
C_e \frac{\partial T_e}{\partial x} &= \nabla(K_e \nabla T_e) - g(T_e - T_l) + S \\
C_l \frac{\partial T_l}{\partial t} &= \nabla(K_l \nabla T_l) + g(T_e - T_l)
\end{aligned}
\tag{1}
$$

In Equation (1), subscript $e$, $l$ represents the electron and lattice subsystems respectively; $T$, $C$, $K$ represents the temperature, specific heat capacity and thermal conductivity of the corresponding system; $g$ is the electron lattice coupling coefficient; and $S$ is the laser light source term.

The experimental results show that under the repetition frequency of 1 KHz, multi-pulse fs laser ablation of copper and nickel also shows an obvious thermal effect, and the three surface topography regions of nickel are larger than those of copper, because the resistivity of nickel is larger than that of copper. According to the Hagen–Ruben relation [26], the absorptivity of metal to 1030 nm laser can be expressed as Equation (2):

$$
A = 365.15\sqrt{\rho/\lambda} = 359.75\sqrt{\rho}
\tag{2}
$$

$\rho$ is the metal resistivity, and $\lambda$ is the laser wavelength. The resistivities of copper and nickel at 20 °C are $1.7 \times 10^{-8}\ \Omega \cdot m$ and $6.8 \times 10^{-8}\ \Omega \cdot m$, respectively, and their densities are $8.96 \times 10^{-3}\ ng/\mu m^3$ and $8.90 \times 10^{-3}\ ng/\mu m^3$, respectively. The resistivity of nickel is higher than that of copper, so its laser absorptivity at 1030 nm wavelength is higher, and the surface topography area after ablation is larger.

Since the thermal effect of metal irradiated by ultrashort laser has a cumulative effect, the plasma dynamics induced by ultrashort laser will also be affected. Figure 7 shows the two-dimensional and three-dimensional ablation height profiles of copper and nickel under the action of 200 energy, 1000 pulses, and different laser pulse widths.

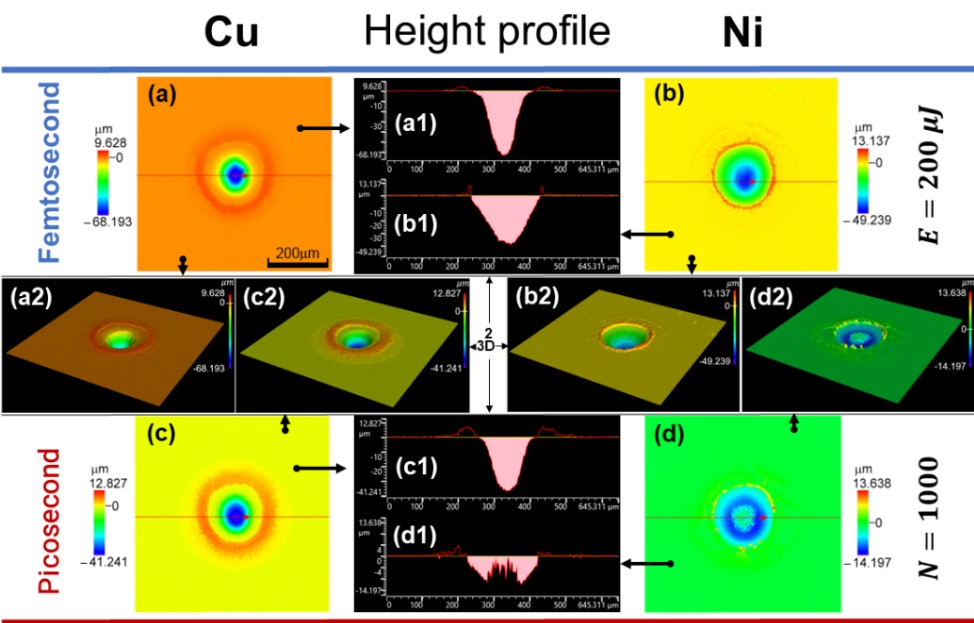

**Figure 7.** Height profile of copper and nickel after laser irradiation with different pulse widths of energy 200 μJ and pulse number of 1000. ((**a,b,c,d**) are the surface ablative morphologies of copper and nickel craters; (**a1,b1,c1,d1**) are 2D ablative height profiles of copper and nickel craters; (**a2,b2,c2,d2**) are 3D ablative height profiles of copper and nickel craters).

Compared with Figure 7(a1,b1,c1,d1), it can be seen that under the same conditions, the ablation depth of copper is greater than that of nickel, and the wall of the crater of the former is smoother, which is consistent with the experimental results of Shaheen et al. [15], and for the same material, the laser ablation depth of fs laser is deeper. At the same time, it can be observed that under the action of 10 ps laser, there is an obvious bulge in the height profile of the bottom of the crater of nickel under the above experimental conditions.

Figure 8 reflects the trend of ablation depth of copper and nickel under the action of ps and fs laser with the increase in pulse number. As can be seen from the diagram, under all the experimental conditions, the ablation depth of copper is basically greater than that of nickel, and the ablation depth varies more linearly with the number of pulses. The ablation depth of fs laser is greater than that of ps laser, and the slope of the whole curve tends to decrease with the increase in pulse number. Combined with the research of Jiang Lan and others, it can be known that the plasma induced by multipulse fs laser irradiation will be affected by pulse number. The result of plasma accumulation similar to the thermal effect will shield part of the laser energy that reaches the surface of the metal material, resulting in a slow trend of ablation depth increasing with the number of pulses. In addition, the restriction of the ablation hole and the roughness of the crater wall will affect the ablation depth, while under the working condition of high repetition frequency, the duty cycle of fs laser is smaller than that of ps laser at the same time, and the plasma shielding effect will decrease, so it will lead to deeper ablation depth of fs laser.

The three-dimensional ablation height profile of the ablation crater can be reconstructed by a confocal microscope, as shown in Figure 7(a2,b2,c2,d2), and its volume is measured using an analytical application (measurement accuracy ±2%). The densities of copper and nickel are $8.96 \times 10^{-3}$ ng/μm$^3$ and $8.90 \times 10^{-3}$ ng/μm$^3$, and the change of ablation mass calculated by the volume method is shown in Figure 9. It can be seen that the ablation mass of copper and nickel ablated by multipulse ultrafast laser is greater than that of copper under the same working conditions. Although the ablation depth of copper is larger, we know that the surface topography area of nickel is larger in the analysis of Section 3.3. The ablation mass of nickel is larger. However, under the condition of 10 ps−200 μJ, the law of change is different, which is also reflected in Figures 7 and 8.

Under this condition, the remelting phenomenon of nickel appears as shown in Figure 9, forming a dense columnar bulge on the surface, which affects the laser absorption, which is different from the flat ablation crater of copper.

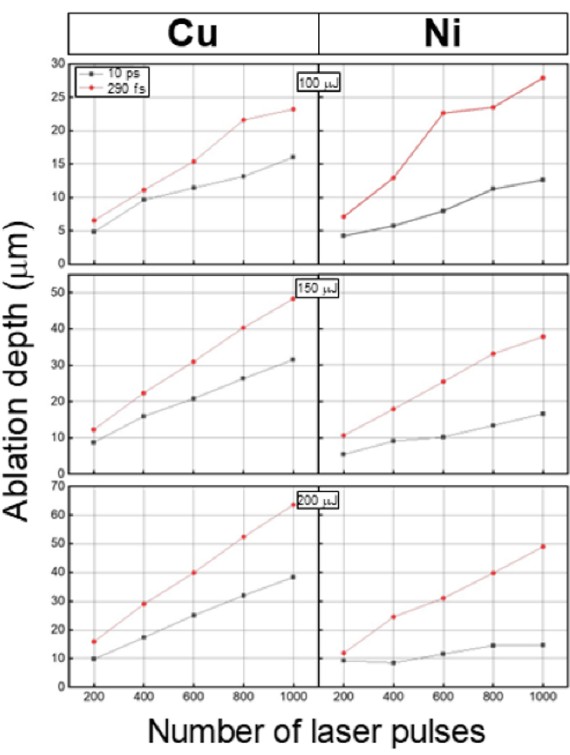

**Figure 8.** Variation trends of copper and nickel ablation depth under the action of ps and fs lasers.

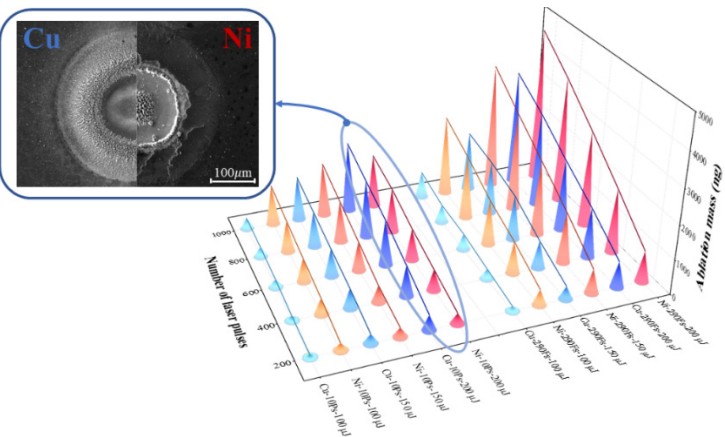

**Figure 9.** Variation trends in ablation mass of copper and nickel.

Figure 8 shows that the ablation depth curve of nickel changes slowly under ps laser ablation, which is actually caused by the remelting phenomenon that affects the measurement of volume. This phenomenon is most obvious at the energy of 200 μJ (Figure 9), resulting in the ablation mass of nickel being lower than that of copper under the same conditions. In the existing studies on the propulsive properties of laser-ablated metals [27], the curves of the single-pulse impulse with laser power density for copper and nickel are close to the same for the same operating conditions in the range of laser power

density $1.5 \times 10^{10} - 3.5 \times 10^{10}$ W/cm$^2$. Equation (3) was used to calculate the specific impulse ($I_{sp}$(s)) of laser ablation micropropulsion [28]

$$I_{sp} = \frac{I}{\Delta mg} \tag{3}$$

where $I$ (kg · m/s) is laser ablation of single-pulse impulse, $\Delta m$ (kg) is the single-pulse ablation mass, and $g$ (N/kg) is the acceleration of gravity. According to the above equation, we can conclude that when the single-pulse ablation mass of copper is smaller, the specific impulse of using copper as a propellant is larger than that of nickel.

### 4. Conclusions

In order to verify the concept of laser ablation to advance the one-dimensional work-piece supply, we performed ablation experiments on surface-polished copper and nickel with submillimeter spots of different energies and pulse numbers (>100) using ultrafast lasers of two pulse widths. The results show that grain refinement occurs in the HAZ of copper after multipulse ultrafast laser ablation, and there will be obvious HAZ on the metal surface after FLMA and PLMA. Compared with copper, the HAZ of nickel is about 5–10% higher than that of copper; The ablation depth of copper is deeper than that of nickel. Under the same conditions, the ablation depth of fs laser is deeper and the ablation crater is more flat; After fs laser ablation, the ablation mass of nickel is greater, but under ps laser ablation, a dense remelting bulge will be formed in the center of the ablation crater of nickel, which affects the laser absorption and volume measurement.

By analyzing the experimental results, ultrafast laser submillimeter spot ablation of copper and nickel, the flatness of the craters of copper is higher than that of nickel, and the variation pattern of the ablation depth of copper with the number of pulses is also more linear, while the ablation mass of copper is smaller than that of nickel, and the specific impulse performance of laser ablation is better than that of nickel. Therefore, copper is more suitable for laser ablation advancement in the one-dimensional work supply mode.

**Author Contributions:** Conceptualization, J.Y., B.D. and Y.W.; methodology, J.Y. and M.L.; software, M.L.; validation, H.G. and C.Y.; formal analysis, M.L. and J.Y.; investigation, L.L. and M.L.; resources, J.Y.; data curation, J.Y. and M.L.; writing—original draft preparation, M.L. and H.G.; writing—review and editing, M.L.; visualization, M.L.; supervision, J.Y. and L.L.; project administration, L.L.; funding acquisition, J.Y. All authors have read and agreed to the published version of the manuscript.

**Funding:** This research received no external funding.

**Institutional Review Board Statement:** Not applicable.

**Informed Consent Statement:** Not applicable.

**Data Availability Statement:** The data presented in this manuscript can be obtained from the corresponding author upon request.

**Acknowledgments:** Thanks to B.D., the Technical Institute of Physics and Chemistry, Chinese Academy of Sciences, for her guidance on the SEM measurement and analysis.

**Conflicts of Interest:** The authors declare no conflict of interest.

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
