# Peer review of "Comparison of Submillimeter Spot Ablation of Copper and Nickel by Multipulse Picosecond and Femtosecond Laser"

_metals, doi:10.3390/met12111971_

Round 1

Reviewer 1 Report

Attractive topic for investigation and very interesting results. Research methods chosen correctly and results of research described clearly. Necessary to continue research in the field of development of technology.

Author Response

Thank you very much for your great valuable times and efforts in kindly helping review our manuscript entitled “Comparison of submillimeter spot ablation of Copper and Nickel by multi-pulse picosecond and femtosecond laser”. (Manuscript ID: metals-1943561, by Li et al). We have made changes to the manuscript in response to your suggestions, and the details of the response are attached.

Reviewer 2 Report

In this work, Li et al proposed a large spot laser ablation with one-dimensional supply mode for laser ablation micropropulsion. They performed ablation experiments on surface-polished copper and nickel with sub-millimeter spots of different energies and pulse numbers (>100) using ultrafast lasers of two pulse widths to verify the concept of laser ablation to advance the one-dimensional workpiece supply. This work is interesting and has potential for publication in the journal. However, some revisions are required to further improve the manuscript quality by considering the following comments.

1.     Introduction: the recent progress and the research problem need further clearer clarification. In addition, Figure 1 should be removed.

2.     Section 2. Experimental equipment and materials: all verbs that are used to describe experiments should be in past tense.

3.     The current discussions are most with results. Therefore, the Section Results and Section Discussion are recommended to be merged into one Section Results and Discussion.

4.     Figure 5: please re-draw professional scale bars because the current ones are hard to read.

5.     Table 2 should be removed because they can be easily described by a short sentence.

6.     Section Conclusions: the authors stated that “The results show that grain refinement occurs in the HAZ of copper after multi pulse ultrafast laser ablation, and there will be obvious HAZ on the metal surface after FLMA and PLMA.” However, there is no results on grain refinement in HAZ. Please add some SEM results to show grain refinement in HAZ.  

Author Response

(The authors gave the same response as above.)

Reviewer 3 Report

The paper entitled “Comparison of submillimeter spot ablation of Copper and Nickel by multi-pulse picosecond and femtosecond laseraddresses the use of ultrafast lasers with two pulse widths of 290 fs and 10 ps to study the comparison of ablation of copper and nickel with different energies and laser pulse numbers at a fixed repetition rate. The paper has scientific interest and originality in its technical content to merit publication. However, it presents some errors that can be improved. It presents a lack of care in its elaboration, namely in the execution of graphics and legends. Grammar has to be considerably improved and has a very basic bibliography. I am sending a copy of the manuscript (pdf) in which all the suggestions/corrections are highlighted.

Author Response

(The authors gave the same response as above.)

Round 2

Reviewer 3 Report

After the implementation of the measures taking into account the improvement of the paper it is in a condition to be accepted for publication